# Aberrant induction of p19$^{Arf}$-mediated cellular senescence contributes to neurodevelopmental defects

**Muriel Rhinn**[1,2,3,4]*, **Irene Zapata-Bodalo**[1,2,3,4], **Annabelle Klein**[1,2,3,4], **Jean-Luc Plassat**[1,2,3,4], **Tania Knauer-Meyer**[1,2,3,4], **William M. Keyes**[1,2,3,4]*

**1** Institut de Génétique et de Biologie Moléculaire et Cellulaire (IGBMC), Illkirch, France, **2** UMR7104, Centre National de la Recherche Scientifique (CNRS), Illkirch, France, **3** U1258, Institut National de la Santé et de la Recherche Médicale (INSERM), Illkirch, France, **4** Université de Strasbourg, IGBMC UMR 7104- UMR-S 1258, Illkirch, France

* rhinn@igbmc.fr (MR); bill.keyes@igbmc.fr (WMK)

**Data Availability Statement:** RNA sequencing data is available at GEO (GSE175680). All other relevant data are within the paper.

## Abstract

Valproic acid (VPA) is a widely prescribed drug to treat epilepsy, bipolar disorder, and migraine. If taken during pregnancy, however, exposure to the developing embryo can cause birth defects, cognitive impairment, and autism spectrum disorder. How VPA causes these developmental defects remains unknown. We used embryonic mice and human organoids to model key features of VPA drug exposure, including exencephaly, microcephaly, and spinal defects. In the malformed tissues, in which neurogenesis is defective, we find pronounced induction of cellular senescence in the neuroepithelial (NE) cells. Critically, through genetic and functional studies, we identified p19$^{Arf}$ as the instrumental mediator of senescence and microcephaly, but, surprisingly, not exencephaly and spinal defects. Together, these findings demonstrate that misregulated senescence in NE cells can contribute to developmental defects.

## Introduction

Cellular senescence is a form of permanent cell cycle arrest induced in response to a variety of stimuli. Senescence arrest is mediated by activation of cell cycle inhibitors including p21, p16$^{Ink4a}$, and p19$^{Arf}$ [1–3]. In addition, the arrested cells are highly secretory, producing a complex cocktail of cytokines, growth factors, extracellular matrix, and other proteins, collectively known as the senescence-associated secretory phenotype (SASP). This can exert significant functional effects on the microenvironment, prominently including the activation and recruitment of immune cells to remove the senescent population. However, the SASP can also exert other effects including promoting cell proliferation, angiogenesis, and epithelial–mesenchymal transition (EMT), in addition to cell plasticity and stemness [4–6]. Although senescence is mostly associated with aging and disease, other studies have shown how senescent cells can have beneficial functions in various settings including embryonic development, tissue repair and regeneration, and tumor suppression and reprogramming [1,2,7,8]. Therefore, the

**Funding:** This work was supported by grants from La Fondation pour la Recherche Medicale (FRM) (AJE20160635985), Fondation ARC pour la Recherche sur le Cancer (PJA20181208104), IDEX Attractivité - University of Strasbourg (IDEX2017), La Fondation Schlumberger pour l'Education et la Recherche FSER 19 (Year 2018)/FRM, Agence Nationale de la Recherche (ANR) (ANR-19-CE13-0023-03) and Ligue Contre le Cancer (all to W.M. K.). I.Z.B. was supported by a 4th year fellowship from the Fondation ARC pour la Recherche sur le Cancer, and a PhD fellowship from INSERM and Conseil Regional Grand-Est. A.K. was supported by fellowship from Eur IMCBiO. The work was also supported by an institutional grant to the IGBMC, ANR-10-LABX-0030-INRT, a French State fund managed by the Agence Nationale de la Recherche under the frame program Investissements d'Avenir ANR-10-IDEX-0002-02. Sequencing was performed by the GenomEast platform, a member of the "France Génomique" consortium (ANR-10-INBS-0009). The funders had no role in study design, data collection and analysis, decision to publish, or preparation of the manuscript.

**Competing interests:** The authors have declared that no competing interests exist.

**Abbreviations:** AER, apical ectodermal ridge; ASD, autism spectrum disorder; BP, basal progenitor; E, embryonic day; EMT, epithelial–mesenchymal transition; HDACi, histone deacetylase inhibitor; NE, neuroepithelial; PHH3, phospho-histone H3; qRT-PCR, quantitative real-time PCR; RG, radial glial; RNA-seq, RNA sequencing; SASP, senescence-associated secretory phenotype; SA-β-gal, senescence-associated beta-galactosidase; TUNEL, TdT-mediated dUTP nick end-labeling; VPA, valproic acid; VZ, ventricular zone.

current view is that timely, controlled, and efficiently cleared senescent cells can have beneficial effects on tissue development and regeneration. However, when there is mistimed or chronic induction of senescence, then this contributes to aging and disease including neurodegenerative disease, fibrosis, and arthritis [2,3,9].

During embryonic development, cells exhibiting features of senescence are detected in precise areas and at critical stages of development, including in the apical ectodermal ridge (AER) of the limb, the hindbrain roofplate, the mesonephros, and the endolymphatic sac [10,11]. Here, it is thought that the controlled induction of senescence contributes to cell fate patterning and tissue development, while the efficient removal of these cells aids in tissue remodeling [1,2,12]. In the embryo, senescence is mediated by p21, but appears not to involve the induction of p16$^{Ink4a}$ and p19$^{Arf}$, which are both expressed from the *Cdkn2a* gene (Ink4a/Arf locus). Indeed, in the embryo, this locus is epigenetically silenced and becomes active in adult life in response to oncogene expression or the aging process [13–15]. Therefore, as mistimed induction of senescence is linked with many adult diseases, we wanted to explore whether aberrant senescence might be implicated in developmental disease.

As a first model to investigate such a possible association, we investigated embryonic exposure to valproic acid (VPA). This drug is widely used to treat a number of illnesses, including epilepsy and bipolar disorder. However, since its initial use, there have been many thousands of cases of women taking VPA during pregnancy, subsequently giving birth to children with birth defects [16–18]. In many cases, these were inadequately counseled about the associated risks, and drug use during pregnancy has continued. Common associated congenital malformations include spina bifida, facial alterations, and heart malformation, with additional risk of limb defects, smaller head size (microcephaly), cleft palate, and more, with higher doses associated with increased risk [16–18]. However, the most widespread consequences of VPA exposure are cognitive impairment and autism spectrum disorder (ASD), which occur in 30% to 40% of exposed infants, and which can occur without any major physical deformity [16,19–21].

The connection between VPA exposure and birth defects has been aided significantly by studies in rodent and primate models, leading to the hypothesis that cognitive defects arise from disruption of early neurodevelopment, around the stage of neural tube closure [21–24]. During this period (approximately embryonic day (E) 8.5 to E9.5 in mice), the early neuroepithelium amplifies, bends, and closes to generate the neural tube, which is lined by neuroepithelial (NE) cells. During neural tube closure, the NE cells divide symmetrically to self-renew and expand [25]. With the onset of neurogenesis, they differentiate into radial glial (RG) cells, which then undergo symmetric proliferative divisions to amplify their pool in the ventricular zone (VZ) of the neuroepithelium [26]. As development proceeds, they transition to asymmetric neurogenic divisions to produce cortical neurons directly or indirectly by amplifying progenitors including the basal progenitors (BPs) [26–28]. These steps must be tightly coordinated, and any perturbation of NE or progenitor function may have consequences on later cortical neuron development that could contribute to microcephaly and other neurodevelopmental disorders including cognitive impairment and ASD.

The molecular mechanisms by which VPA perturbs development are mostly unknown, but likely result from its function as a histone deacetylase inhibitor (HDACi) [29]. Interestingly, in this capacity, VPA is also broadly used in cancer therapy and has been shown to induce cellular senescence in certain settings, through direct activation of key senescence mediators including p21, p16$^{Ink4a}$, and p19$^{Arf}$ [30]. Given these associations, we investigated whether aberrant activation of senescence by VPA exposure might contribute to the associated developmental defects.

## Results

### Valproic acid induces exencephaly, microcephaly, and spinal cord defects in mice

Drawing from earlier VPA exposure studies in mice [23,31], we established a time-course paradigm for assessing acute and developmental phenotypes caused by VPA during embryonic development (see experimental scheme Fig 1A). Although it has been shown that acute dosing of mice can model many key features of drug exposure in humans, mice have high drug tolerance and clearance capacity, and as such, comparatively higher doses are used to model exposure. Also, although in humans VPA causes spina bifida, a posterior neural tube closure defect where part of the spinal cord and nerves are exposed, in mice, exencephaly, a defect of anterior neural tube closure where the brain is located outside of the skull, has been noted [32]. Here, we first analyzed E13.5 embryos from pregnant female mice that had been dosed 3 times around E8. As previously observed, we identified prominent and recurrent defects, such as exencephaly (Fig 1B). However, we also observed that a large proportion of the mice displayed a small brain phenotype resembling microcephaly, a finding which was previously underestimated in mice (Fig 1B). Next, we analyzed VPA-exposed embryos at earlier developmental

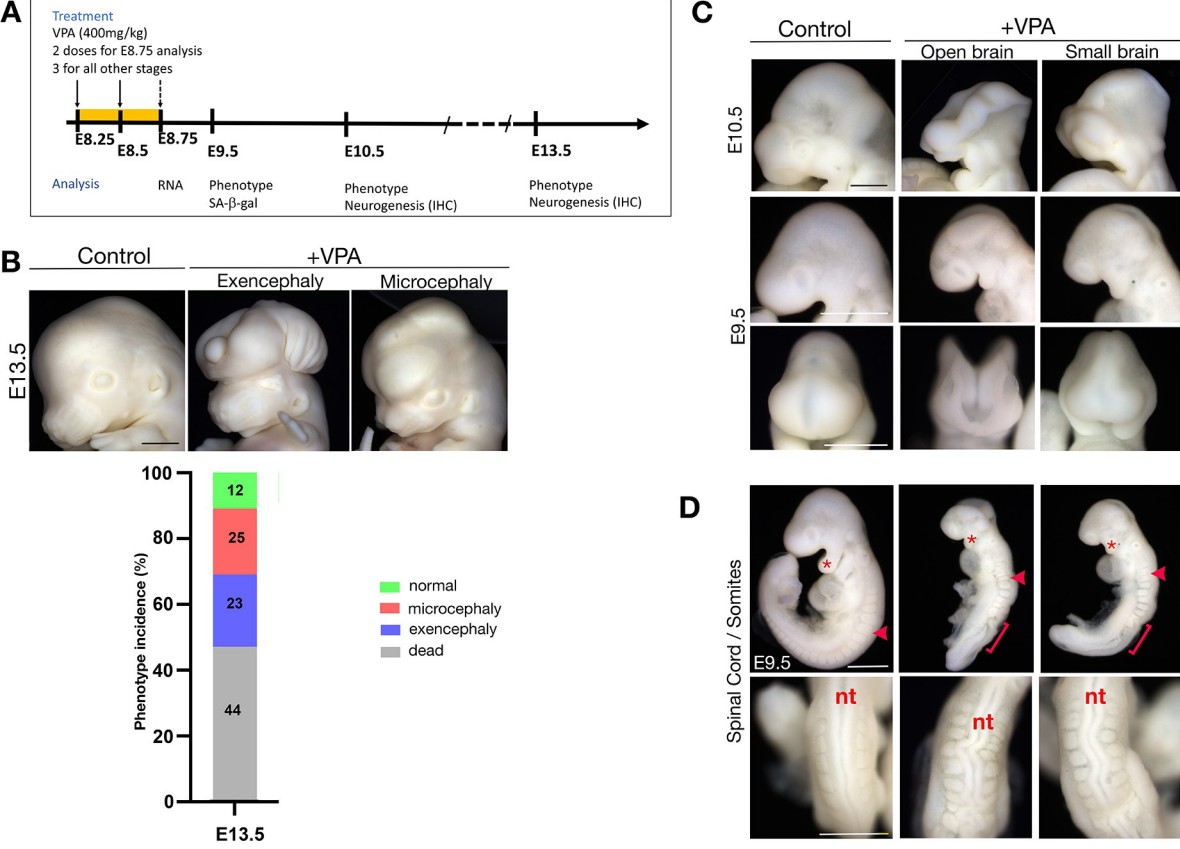

**Fig 1. VPA treatment induces developmental defects, including exencephaly, microcephaly, and abnormal spinal cord development.** **(A)** Schematic of experimental treatment of mice with VPA and timeline of analysis. **(B)** Top: embryonic head phenotypes in CD1 mice at E13.5 resulting from VPA exposure. Scale bar, 1 mm. Bottom: phenotype incidence at E13.5 (*n* = 45 embryos from 4 litters). **(C)** Embryonic head phenotypes in CD1 mice at E10.5 and E9.5. Scale bar, 500 μm. **(D)** Lateral views (top) and dorsal views (bottom) of control and VPA-treated embryos dissected at E9.5, illustrating the pronounced curve in the nt and abnormally shaped somites observed (arrowhead and brackets), and hypoplastic pharyngeal arches (asterisks). Scale bar, 500 μm. E, embryonic day; IHC, Immunohistochemistry; nt, neural tube; VPA, valproic acid.

stages and could visually distinguish these same phenotypes at E10.5 (Fig 1C). When we examined even earlier embryos at E9.5, they also presented with characteristic phenotypes, but at these early stages, care needs to be taken with regard to potential differences in developmental timing. In general, these embryos frequently presented with open neural tube (approximately 29%) and/or smaller brains (approximately 39%) (Fig 1C), suggesting that these may ultimately give rise to, respectively, the exencephaly and microcephaly phenotypes observed at later stages. Furthermore, at these earlier stages, additional deformations were obvious, including somite absence, fusion or gross misalignment (Fig 1D), kinked neural tubes, and hypoplastic pharyngeal arches. Critically, quantitative measurements showed that VPA-treated embryos, just like controls, had all turned, but were significantly shorter in length, and had fewer quantifiable somites as a result of the malformations (S1 Fig). This analysis uncovers distinct separate responses to VPA that were not previously characterized and demonstrate that VPA can cause early phenotypic changes during mouse brain development that recapitulate features of VPA exposure in humans.

## Valproic acid induces ectopic senescence in neuroepithelial cells

Next, we investigated whether cellular senescence was a feature in VPA-exposed mouse embryos. First, we performed wholemount staining to assess for activity of the senescence marker senescence-associated beta-galactosidase (SA-β-gal) on E9.5 control or VPA-exposed embryos presenting with the open-brain or small-brain phenotypes. We found that ectopic SA-β-gal activity was prominent in the forebrain and hindbrain in both open-brain and small-brain embryos (Fig 2A, arrow). Notably, this ectopic staining was absent in both the spinal cord and the malformed somites. When we sectioned the embryos, we found that SA-β-gal activity was localized in the NE cells, the embryonic precursors of neurons and glia in the brain (Fig 2B). Interestingly, the SA-β-gal staining was predominantly localized at the apical border of the NE cells. We next assessed proliferation in these cells to confirm their senescent status. Measuring EdU incorporation, we confirmed that VPA-exposed mouse embryos had a significant decrease in staining throughout the forebrain, which was noticeably reduced in the apical borders (Figs 2C, S2A, and S2B). To confirm this, we also performed anti-phospho-histone H3 (PHH3) staining, which labels apical NE cell proliferation, and which again showed a significant reduction in proliferation in the NE cells of VPA-treated embryos (S2C Fig). Next, we assessed cell death levels by wholemount TUNEL staining. Here, the VPA-exposed embryos had a visible increase of cell death in the forebrain regions, while, as expected, both control and VPA-exposed embryos had cell death at the neural fold tips. However, when sectioned, we did not detect any cell death in the NE cells where the senescence staining was located, further supporting that VPA induces senescence in the NE cells (S3 Fig). Finally, we dissected the forebrain and midbrain regions from wild-type or VPA-exposed small-brain embryos at E8.75 and performed quantitative real-time PCR (qRT-PCR) for senescence genes, including cell cycle inhibitors and secreted components of the SASP. We found that $p21$, $p19^{Arf}$, and $p16^{Ink4a}$ and the SASP genes $IL6$, $IL1a$, $IL1b$, and $Pai1$ were strongly induced in VPA-exposed embryos (Fig 2D). Together, these data uncover that VPA induces ectopic senescence in NE cells during developmental neurogenesis.

## Neural differentiation is reduced by valproic acid exposure

To investigate the potential impact of such aberrant senescence on later cortical development, we analyzed telencephalic corticogenesis at subsequent developmental stages. NE cells undergo differentiation into progenitors, which will then give rise to neurons and glia. When we performed immunostaining in small-brain embryos for the neural progenitor markers Pax6

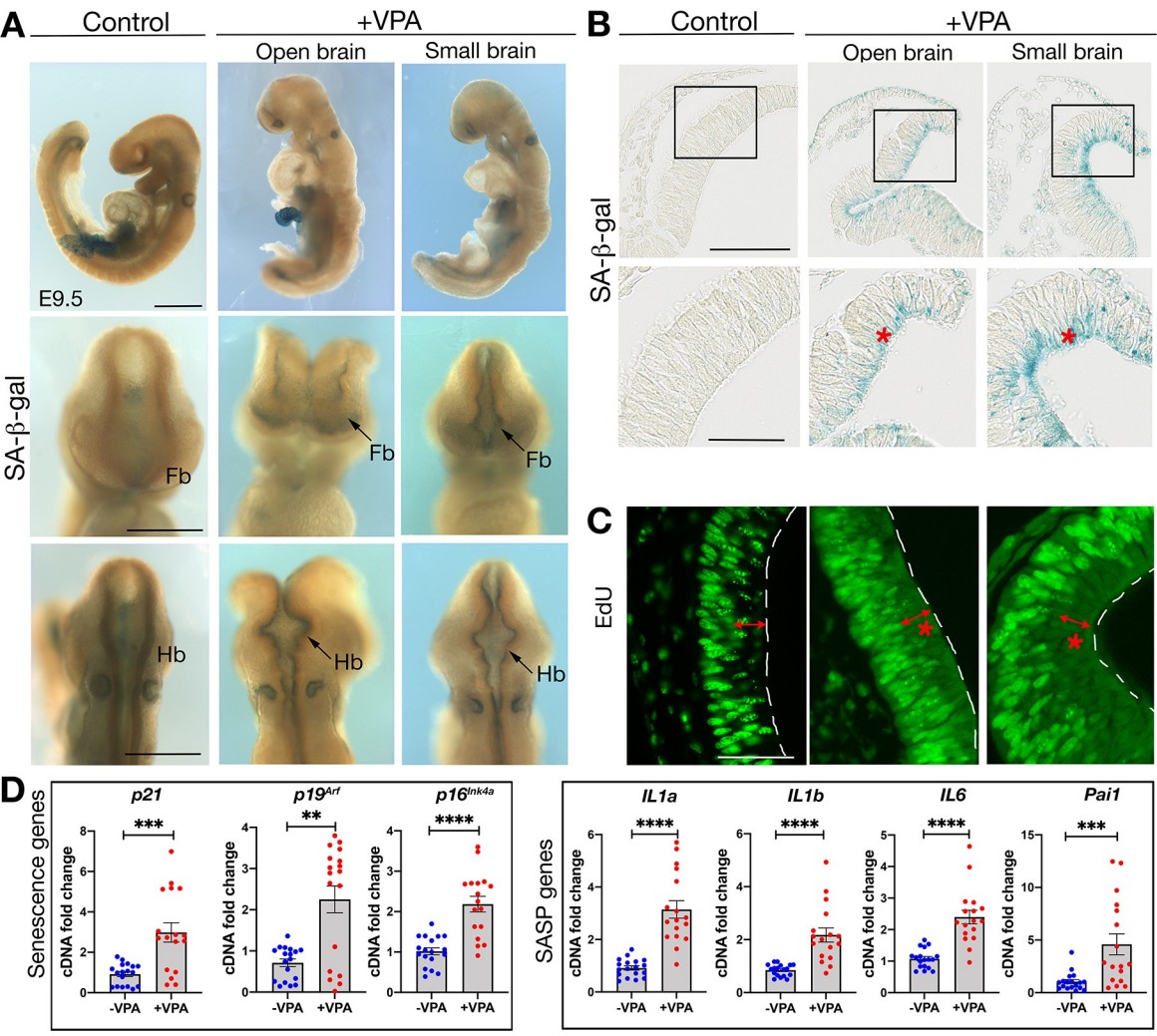

**Fig 2. Senescence is induced in the forebrain and hindbrain neuroepithelium of VPA-treated embryos. (A)** Whole mount SA-β-gal staining in control and VPA-treated embryos at E9.5 (*n* = 18 embryos from 7 litters). Top row, lateral views. Scale bar, 500 μm. Middle row, frontal views and bottom row, dorsal views. Scale bars, 50 μm. Fb, forebrain. Hb, Hindbrain. **(B)** Sections through whole mount SA-β-gal stained forebrains (scale bar, 100 μm). Box shows the region imaged in lower panel (scale bar, 50 μm). Red asterisks highlight senescent cells. (*n* = 8 embryos from 4 litters). **(C)** EdU incorporation in NE cells. Red asterisks indicate location of senescent cells (*n* = 6 embryos from 5 litters), and the double arrows highlight the apical zone. White dashed lines indicate apical surface of the neural tube. EdU, 5-ethynyl-2′-deoxyuridine. Scale bar, 50 μm. **(D)** qRT-PCR analysis on E8.75 forebrain + midbrain, for senescence markers (p21, p19$^{Arf}$, and p16$^{Ink4a}$) and SASP genes (IL1a, IL1b, IL6, and Pai1) (*n* = 17 to 18 embryos from 3 different litters). Data bars represent mean ± SEM. Mann–Whitney test: $^{**}p \leq 0.01$, $^{***}p \leq 0.001$, and $^{****}p \leq 0.0001$. The data underlying this figure can be found in S1 Data. E, embryonic day; NE, neuroepithelial; qRT-PCR, quantitative real-time PCR; SA-β-gal, senescence-associated beta-galactosidase; VPA, valproic acid.

(apical progenitors) and Tbr2 (intermediate progenitors), and for the neuronal differentiation marker Tuj1, we found a significant decrease in progenitors and neurons in VPA-exposed embryos at E10.5 (S4 Fig) and E13.5 (Fig 3). Overall, these data associate early aberrant senescence in NE cells of the embryo with decreased neurogenesis and impaired cortical development.

## Human cerebral organoids exhibit senescence in response to valproic acid treatment

We next sought to assess if VPA exposure might similarly induce senescence in human NE cells and used cerebral organoids to investigate this possibility. We grew human organoids as

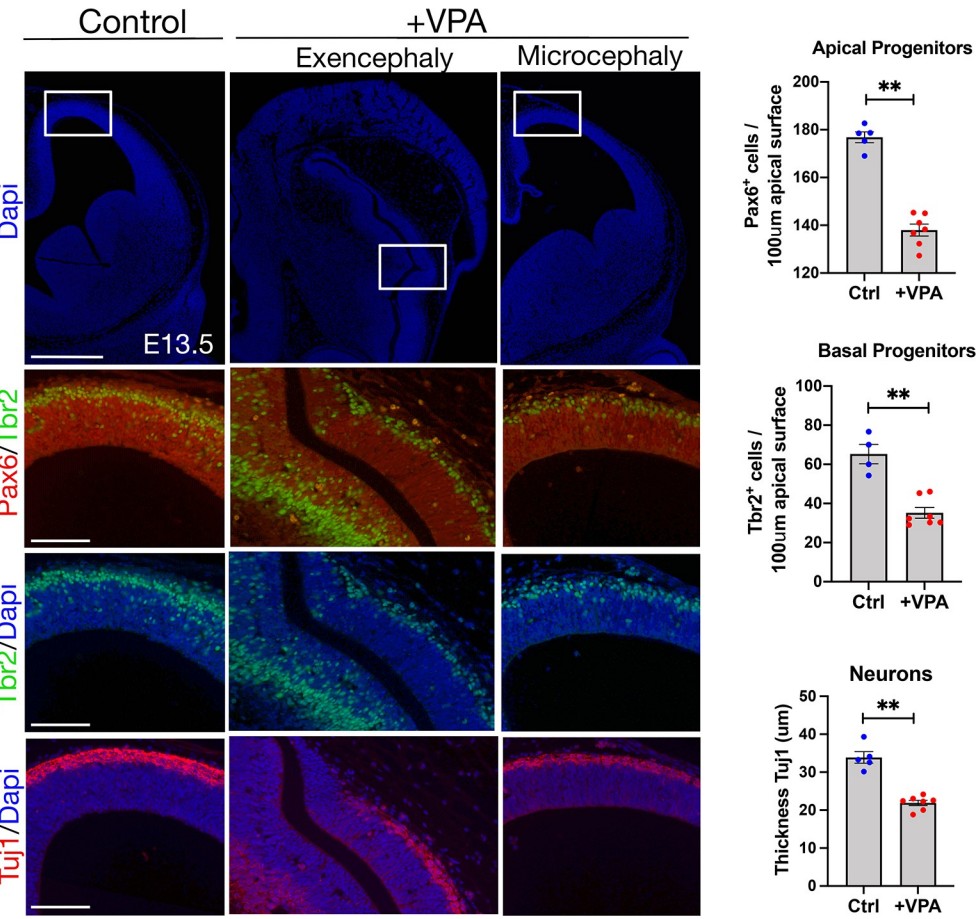

**Fig 3. VPA treatment and senescence induction is associated with decreased neurogenesis.** Immunostaining for Pax6, Tbr2, and Tuj1 on cortical sections (coronal) of E13.5 embryos. Box highlights the region in lower images. Scale bar, 500 μm (top row), 100 μm (rest). Quantification of Pax6 and Tbr2 positive progenitors or the thickness of the neuronal layer in the microcephalic cortical vesicles (for each condition, 5 embryos from at least 4 different mothers were analyzed). Data bars represent mean ± SEM. Mann–Whitney test: $^{**}p \leq 0.01$. The data underlying this figure can be found in S1 Data. E, embryonic day; VPA, valproic acid.

previously described [33] and exposed these to different concentrations of VPA at time points equivalent to the same developmental stages in mouse. Specifically, we treated cultures with 1 to 2 mM VPA from day 18 to 25 and analyzed the organoids upon VPA removal at day 25, or allowed the organoids to develop until day 42, when neuronal differentiation could be assessed (Fig 4A).

Exposure to VPA caused a significant decrease in organoid growth that persisted after drug removal (Fig 4B). As in mice, we assessed cortical neurogenesis in VPA-treated organoids and found a significant reduction in neural rosette size and progenitor number, as measured by Pax6 and Sox1/Tbr2 staining, respectively (Figs 4C and S5), and impaired differentiation of neurons, as measured by Tuj1 (Fig 4C). When we assessed senescence using wholemount SA-β-gal staining, we detected a strong induction in the organoids following VPA treatment, which upon sectioning was found to be present specifically in the NE cells (Fig 4D). Proliferation was also decreased in these cells, as measured by anti-PHH3 staining (Fig 4E). Of course, while it may be considered that rosette size is smaller owing to the decreased total organoid size, we believe that the reduction in rosette size is likely a determinant of the overall size

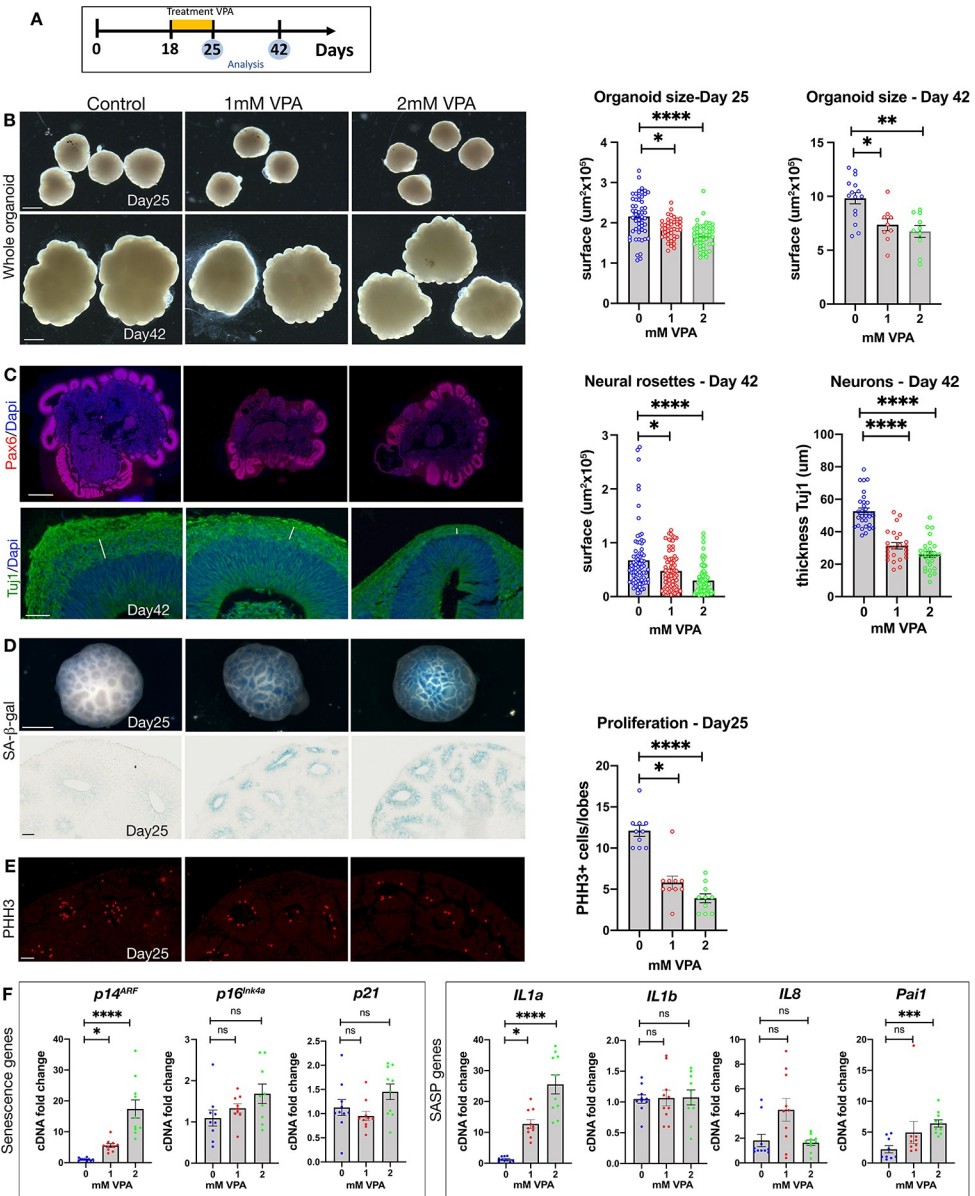

**Fig 4. Human cerebral organoids treated with VPA show a decreased size, impaired neurogenesis, and induction of senescence in NE cells. (A)** Schematic for organoid cultures experiments. **(B)** Left: Bright field images of cerebral organoids at days 25 and day 42. Scale bar, 1 mm. Right: organoid size ($\mu m^2$) at day 25 ($n$ = 52 (Control), 41 (1 mM VPA), 45 (2 mM VPA), 4 independent experiments) and day 42 ($n$ = 15 (Control), 9 (1 mM VPA), 10 (2 mM VPA), 4 independent experiments). Data bars represent mean ± SEM. Kruskal–Wallis test: $^*p \leq 0.05$, $^{**}p \leq 0.01$ and $^{****}p \leq 0.0001$. **(C)** Left: Immunostaining on sections of control and VPA-treated organoids for Pax6 (red) or Tuj1 (green), counterstained with Dapi (blue). Scale bar, 500 μm (Pax6) and 50 μm (Tuj1). Right: Neural rosette area at day 42 ($n$ = 79 (Control), 76 (1mM VPA), 79 (2 mM VPA), 4 independent experiments), and neuron layer thickness (μm) at day 42 ($n$ = 30 (Control), 24 (1 mM VPA), 28 (2 mM VPA), 4 independent experiments). Data bars represent mean ± SEM. Kruskal–Wallis test: $^*p \leq 0.05$, and $^{****}p \leq 0.0001$. **(D)** Whole mount SA-β-gal staining of day 25 organoids (scale bar, 500 μm). Sections show SA-β-gal staining in the neuroepithelium (scale bar, 50 μm) ($n$ = 5 (Control), 5 (1 mM VPA), 5 (2 mM VPA), 3 independent experiments). **(E)** Left: Immunostaining on sections of control and VPA-treated organoids for PHH3 (red) at day 25. Scale bar, 50 μm. Right: Proliferation quantification at day 25. ($n$ = 10 (Control), 10 (1 mMVPA), 10 (2 mMVPA), 2 independent experiments). Data bars represent mean ± SEM. Kruskal–Wallis test: $^*p \leq 0.05$ and $^{****}p \leq 0.0001$. **(F)** qRT-PCR analysis for senescence markers ($p21$, $p14^{ARF}$, $p16^{INK4A}$) and for SASP genes ($IL1a$, $IL1b$, $IL8$, and $Pai1$) ($n$ = 10 organoids from 4 independent experiments). Data bars represent mean ± SEM. Kruskal–Wallis test: ns, not significant, $^*p \leq 0.05$, $^{***}p \leq 0.001$ and $^{****}p \leq 0.0001$. The data underlying this figure can be found in S1 Data. NE, neuroepithelial; PHH3, phospho-histone H3; SA-β-gal, senescence-associated beta-galactosidase; VPA, valproic acid.

impairment, especially as senescence was detected specifically in the NE cells. Finally, we assessed expression of key senescence mediators by qRT-PCR at day 25. Interestingly, we observed a significant induction of $p14^{ARF}$ (human ortholog of p$19^{Arf}$) and the SASP genes *IL1a* and *Pai1*, but no change in $p16^{INK4A}$ or *p21* expression (Fig 4F).

## p19$^{Arf}$ deficiency rescues senescence and microcephaly induced by valproic acid

Thus far, our experiments uncovered that exposure to VPA causes a pronounced induction of senescence in NE cells that is associated with a marked decrease in proliferation and neurogenesis. However, we wanted to investigate if aberrant senescence is functionally coupled to the observed phenotypes and impaired neurogenesis. To address this, we employed genetic loss of function models deficient in the main senescence mediators *p21*, $p19^{Arf}$, or $p16^{Ink4a}$ and treated pregnant mice, each individually deficient for these genes, with VPA, and assessed the E9.5 embryo phenotypes. Surprisingly, we found that *p21*- and $p16^{Ink4a}$-deficient embryos had no visible improvement in any phenotype (S6 Fig). With regard to $p19^{Arf}$-deficient embryos exposed to VPA, these displayed no rescue of open-brain incidence, nor somite number and spinal curvature defects relative to wild-type mice (S7 Fig). Interestingly, however, they were noticeably improved, with regard to the incidence and/or severity of the small-brain phenotype (Fig 5A and 5B). To validate our observations, we measured the combined forebrain and midbrain area in all embryos. At this early stage (1 day after VPA exposure), we found that the forebrain/midbrain size in $p19^{Arf}$-deficient embryos was significantly larger compared to wild-type VPA-exposed embryos, an effect that was not present in *p21*- and $p16^{Ink4a}$-deficient embryos (Fig 5A). The lessened size reduction $p19^{Arf}$-deficient embryos was also evident with in situ hybridization for the forebrain marker *Six3* (S8 Fig). To assess whether the size difference phenotype correlated with changes in senescence, we again assessed SA-β-gal staining and found that VPA-exposed $p19^{Arf}$-deficient mice had reduced expression in the NE cells relative to VPA-exposed wild-type embryos (Fig 5C). Again, this decrease was not detectable in *p21*- and $p16^{Ink4a}$-deficient embryos (S9 Fig). Furthermore, when assessed by qRT-PCR, $p19^{Arf}$ deficiency was associated with a decrease in $p16^{Ink4a}$ and a reduced SASP response (S10 Fig). In agreement with the results from human organoids, this data points to $p19^{Arf}$ as a mediator of VPA-induced senescence in the embryo.

To further investigate this association and to determine if ectopic p19$^{Arf}$ expression is sufficient to induce senescence and cause developmental defects when aberrantly expressed in the neuroepithelium, we electroporated mouse $p19^{Arf}$ into the NE cells of chick embryo forebrains. In comparison to GFP-control plasmid, we found that $p19^{Arf}$ expression caused a unilateral perturbation of development, decreasing forebrain size, and induced strong ectopic SA-β-gal activity in the NE cells (Figs 5D, 5E and S11). These data demonstrate that aberrant $p19^{Arf}$ expression is sufficient to induce senescence and developmental defects.

Finally, to conclusively demonstrate that aberrant senescence contributes to impaired neurodevelopment, we asked whether p19$^{Arf}$ deficiency would rescue some of the major defects caused by VPA exposure. To answer this question, we measured progenitor and neuronal status during cortical neurogenesis at later stages, when neurodevelopment has progressed further. As p19$^{Arf}$ deficiency only rescued the small-brain phenotype at early stages, here now we analyzed the microcephaly phenotype. As before, wild-type embryos exposed to VPA and examined at E13.5 presented with characteristic features of microcephaly, and with a significant reduction in the number of progenitors and neurons (Fig 5F). Strikingly, however, $p19^{Arf}$-deficient mice were not as susceptible to VPA exposure, and presented with a rescue of the microcephalic features, and significantly increased numbers of progenitors, and increased

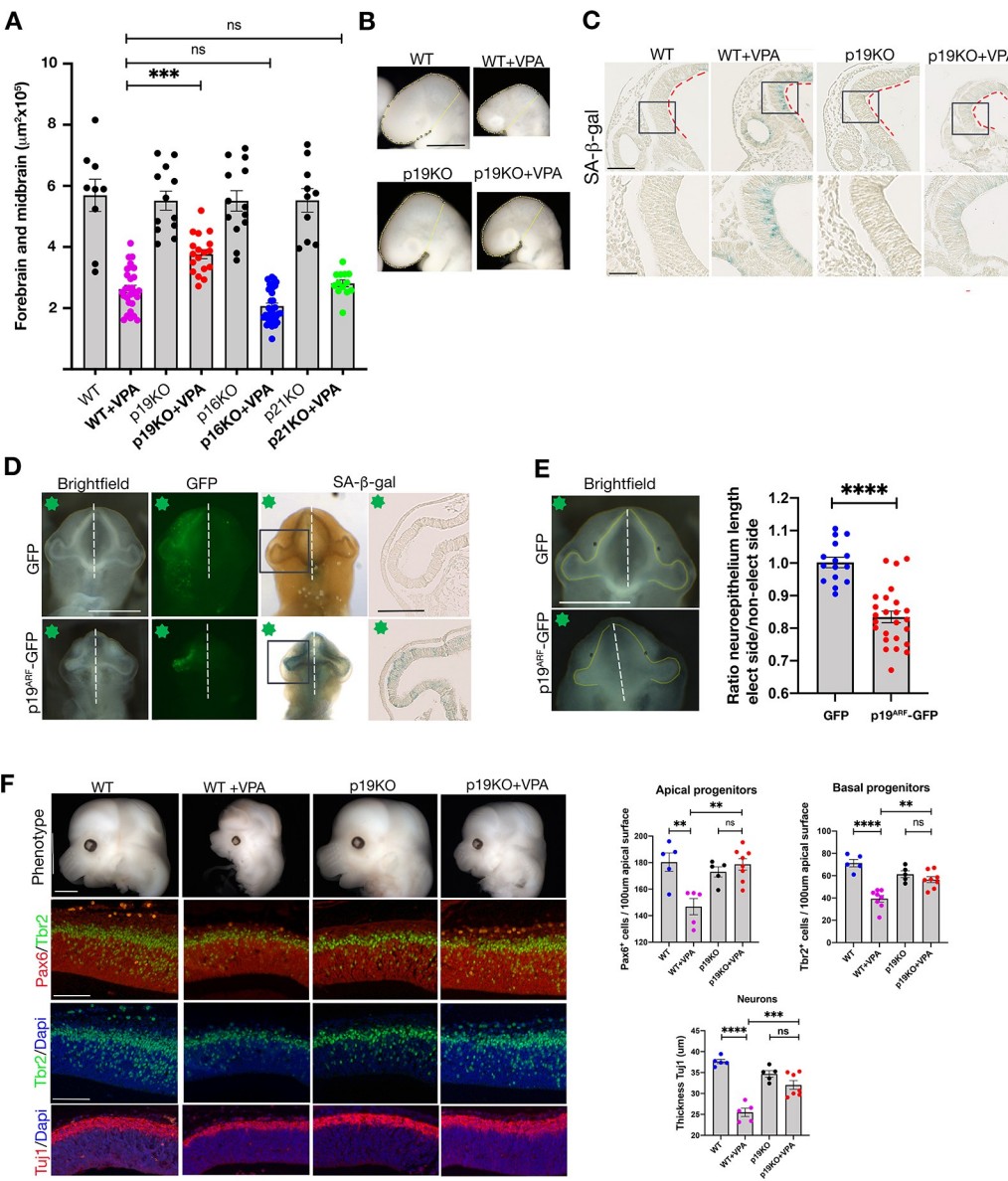

**Fig 5. p19^Arf expression causes senescence and VPA-induced microcephaly. (A)** Graph shows surface area of forebrain and midbrain in each condition, WT, *n* = 9 embryos from 4 litters, WT+VPA, *n* = 29 embryos from 14 litters, p19KO, *n* = 12 embryos from 4 litters, p19KO + VPA, *n* = 18 embryos from 7 litters, p16KO, *n* = 13 embryos from 4 litters, p16KO + VPA, *n* = 34 embryos from 8 litters, p21KO, *n* = 11 embryos from 4 litters, p21KO + VPA, *n* = 13 embryos from 8 litters. Data bars represent mean ± SEM. Kruskal–Wallis test: ns, not significant, $^{**}p \leq 0.01$ and $^{****}p \leq 0.0001$. **(B)** Bright field images of E9.5 embryonic heads, indicating area of the forebrain and midbrain (yellow line). Scale bar, 500 μm. **(C)** Representative brain sections of E9.5 SA-β-gal stained WT or p19KO (scale bar, 100 μm). Box shows the region imaged in lower panel (scale bar, 50 μm). Red dashed lines indicate apical surface of the neural tube. WT, *n* = 5 embryos from 3 litters, WT+VPA, *n* = 6 embryos from 4 litters, p19KO *n* = 5 embryos from 3 litters for p19KO, p19KO + VPA *n* = 9 embryos from 3litters. **(D)** Ventral views of chick embryos at stage HH12, electroporated with a GFP or a p19^Arf-GFP plasmid. Green star indicates electroporated side. Scale bar, 500 μm. Embryos were stained for SA-β-gal activity. Boxes indicate sectioned area of forebrain neuroepithelium shown. Scale bar, 100 μm **(E)** Brightfield embryos with yellow line shows length of neuroepithelium. Scale bar, 500 μm. Graph shows ratio of length of neuroepithelium in electroporated side compared to control side. GFP, *n* = 15 embryos from 4 different electroporations, p19^Arf-GFP, *n* = 25 embryos from 9 different electroporations. Data bars represent mean ± SEM. Unpaired *t* test: $^{****}p \leq 0.0001$. **(F)** Images showing the cortical vesicles from microcephaly embryos. Scale bar, 1 mM. Immunostaining on cortical sections, E.13.5, for Pax6, Tbr2, Tuj1, and counterstained with Dapi. Scale bar, 100 μm. Graphs showing number of Pax6 and Tbr2 positive progenitors or the thickness of the Tuj1 neuronal layer in the cortical vesicles (for each condition, minimum 5 embryos from at least 4 different mothers were analyzed). Data bars represent mean ± SEM. One-way ANOVA plus

Tukey post hoc test: ns, no significant, $^{**}p \leq 0.01$, $^{***}p \leq 0.001$ and $^{****}p \leq 0.0001$. The data underlying this figure can be found in S1 Data. E, embryonic day; KO, knockout; SA-β-gal, senescence-associated beta-galactosidase; VPA, valproic acid; WT, wild-type.

thickness of the neuronal zone relative to wild-type embryos. These experiments conclusively demonstrate that p19$^{Arf}$, in response to VPA, drives a senescence-mediated block in neurogenesis.

## Pathways associated with neurodevelopmental defects are rescued in p19$^{Arf}$-deficient mice

Given that *p19$^{Arf}$* deficiency is protective for early VPA-induced embryonic developmental defects, we wanted to begin to understand the underlying mechanism at a molecular level. To this end, we performed RNA sequencing (RNA-seq) on the forebrain/midbrain region from both wild-type and *p19$^{Arf}$*-deficient embryos, either treated or untreated with VPA. Through phenotype pathway analysis of differentially expressed genes, it was evident that many neuro-developmental and ASD-related phenotypes, including exencephaly and microcephaly, were associated with significantly down-regulated genes in VPA-exposed wild-type mice (Fig 6A). Specifically, these gene signatures were associated with Wnt and Hippo signaling [34] (S12A Fig). In *p19$^{Arf}$*-deficient animals, however, most of these signatures were significantly less affected, confirming our phenotypic observations of the genetic backgrounds (Figs 6A and S12A).

Genetic population studies have identified candidate genes associated with microcephaly and ASD [35,36]. Many of these genes are significantly decreased in both the forebrain and midbrain of VPA-exposed wild-type embryos, including Chd8, Dyrk1a, Fmr1, Cep63, and others. However, most were not restored upon *p19$^{Arf}$* loss (S12B and S12C Fig), suggesting that senescence may be regulated independently or downstream of these specific genes. There-fore, to get a better understanding of how p19$^{Arf}$ might induce these ectopic phenotypes, we analyzed the subset of genes that were significantly down-regulated in VPA-exposed wild-type embryos, but that were not significantly decreased in *p19$^{Arf}$*-deficient embryos (genes depicted in red in Fig 6B). Within this p19$^{Arf}$-dependent gene set, we identified tRNA aminoacylation and tRNA export (Figs 6C and S12D). Interestingly, perturbation of tRNAs or their regulatory mechanisms is linked to microcephaly and neurodevelopmental defects [37]. This suggests that p19$^{Arf}$-mediated senescence and repression of these genes may contribute to microcephaly and cognitive impairment.

## Discussion

Together, these findings demonstrate that aberrantly induced senescence perturbs embryonic development, leading to developmental defects, and advances our understanding of how VPA causes neurodevelopmental disorders.

A major finding of this work is that it makes an exciting functional connection between aberrant cellular senescence and developmental defects. While abnormal induction or chronic accumulation of senescent cells has been linked to many adult and age-related diseases, we demonstrate here a causative role for senescence in neurodevelopmental defects. Interestingly, we identify that the NE cells are the site of senescence induction. As this population of cells is a critical precursor of all mature cell types in the brain, it stands to reason that this is one of the most perturbed population of cells in neurodevelopmental disorders. We demonstrate that induction of senescence in the NE cells correlates with a subsequent impairment in

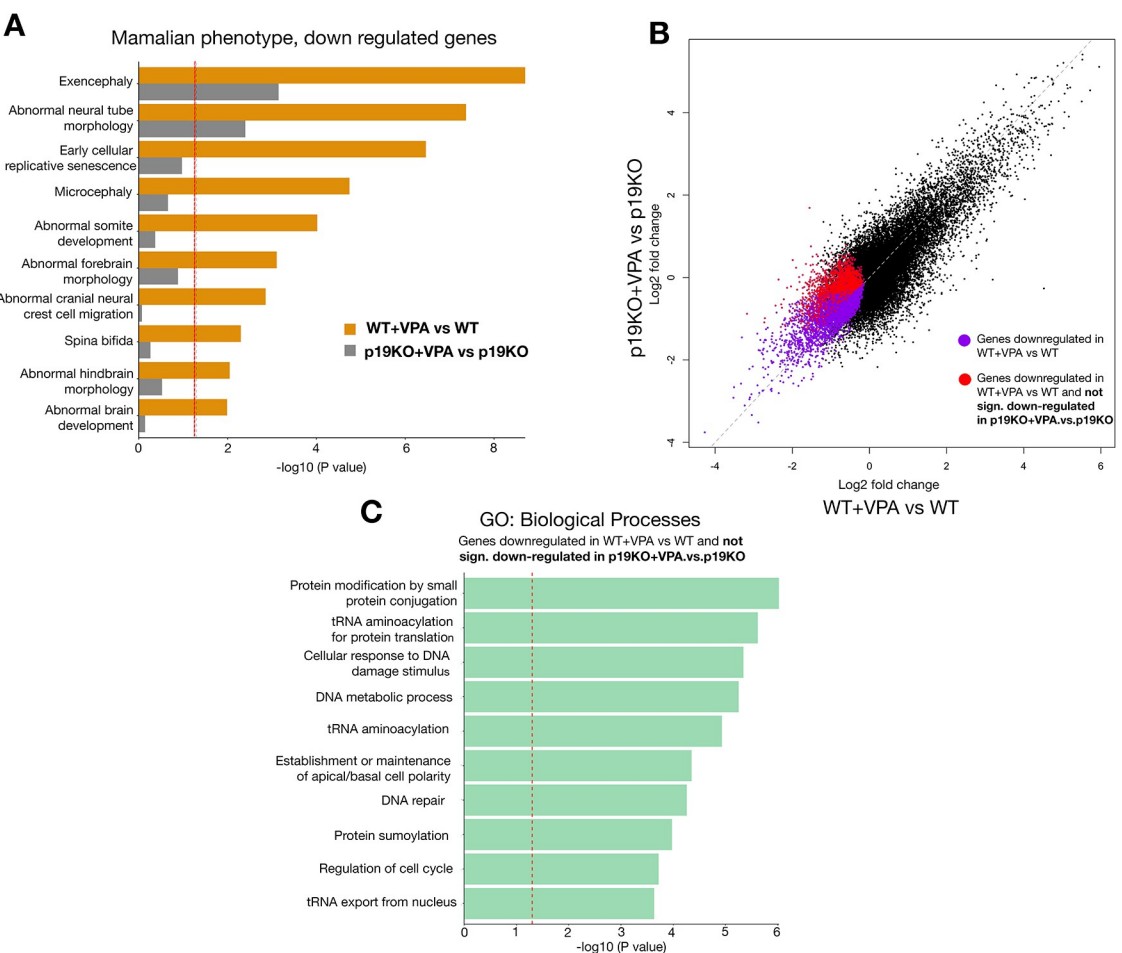

**Fig 6. _p19^Arf_ deficiency rescues VPA-induced gene signatures associated with neurodevelopmental defects. (A)** Selected Mammalian Phenotype pathway analysis terms on the down-regulated genes from RNA-seq of the forebrain and midbrain **(B)** Scatter plot showing mRNA fold changes for the genes in WT+VPA compared to WT, and in p19KO + VPA compared to p19KO. **(C)** GO Biological Process pathway analysis on genes highlighted in E with red dot. The data underlying this figure can be found in S1 Data. KO, knockout; RNA-seq, RNA sequencing; VPA, valproic acid; WT, wild-type.

corticogenesis and neural differentiation, which is rescued in the absence of a key senescence gene. This demonstrates that this induction of senescence effectively blocks the development of the affected NE cells. As the majority of infants with problems associated with VPA exposure have cognitive defects, including developmental delay and ASD, this suggests that senescence in the NE cells could be a significant contributor to these outcomes.

This study also links aberrant senescence in the NE cells with a small-brain phenotype, characteristic of microcephaly. Indeed, microcephaly is a feature of VPA exposure in infants, and the strategy used here in mice of an acute model of VPA exposure, mimics many associated features of VPA exposure in humans [16,23]. Such high-dose acute treatment is necessary to avoid the low penetrance of developmental defects seen in mice. Of course, it is possible that this may exaggerate some of the features found in humans. However, as shown by VPA treatment of human organoids during week-long exposure, the outcome of senescence in NE cells is conserved, correlating with increased expression of _p14^Arf_ and decreased neurogenesis. As affected human embryos are chronically exposed to the drug during development, it is possible that this would cause a lower, but longer incidence of senescence in NE cells or their

derivatives, but which may perturb differentiation in specific areas or at different stages of development, yet without always manifesting as microcephaly. Interestingly, however, there is a strong correlation between microcephaly at birth and lower cognitive ability in ASD patients [38–42], suggesting that further exploration for possible connections between mistimed senescence during development and ASD is warranted.

How VPA causes birth defects has remained unclear, but exposure during the first trimester, around the stages of neural tube closure, is suggested as being critical in driving the phenotypes associated with this drug, and with higher doses associated with increased risk [16,21,24]. Our findings identify that the drug can affect individual embryos differently, causing severe physical defects such as exencephaly in some, while causing different effects such as microcephaly in others. The reasons for this varied response remain unknown, but is likely related to cell type–specific responses. For example, while we did not detect aberrant cell death in the NE cells, apoptosis was apparent on the surface ectoderm after VPA treatment, suggesting that in some cases, VPA-induced cell death could contribute to the phenotypes. Interestingly, our study also identifies early and severe posterior neural tube and somite defects in mice, which may improve as development proceeds, as these were not as severe at E13.5. However, these were not associated with senescence, nor were they rescued in $p19^{Arf}$-deficient mice. Together, this supports that aberrant senescence may be more associated with neurodevelopmental defects, as opposed to major congenital deformations.

It might also be considered surprising that the senescence-induced phenotypes are mediated by p19$^{Arf}$, and not p21 or p16$^{Ink4a}$, the latter of which is often considered a primary mediator of adult and age-associated senescence [1,2,13]. One possibility may be because the *Ink4a/Arf* locus is directly repressed by HDACs, which contributes to the normal silencing of these genes in the embryo. However, HDACis including VPA can directly derepress this locus, and it appears that VPA has preferential ability for activating p19$^{Arf}$ over p16$^{Ink4a}$ [43,44]. In support of this association, we demonstrate that ectopic expression of *p19$^{Arf}$* is sufficient and able to cause senescence, impaired neurogenesis, and developmental defects. Another possibility may relate to the timing and duration of senescence. Interestingly, in senescence induced in cells in culture, p16$^{Ink4a}$ expression often appears later in the program. Perhaps, in this case in the embryo, the senescent cells are transiently induced following VPA exposure and are ultimately cleared before expression of p16$^{Ink4a}$ can manifest. Therefore, it is possible that p16$^{Ink4a}$, or even misexpressed p21, could contribute to other developmental defects.

An outstanding question as to why there is such a restricted pattern of senescence induced in the embryo by VPA is likely related to the pattern of expression of HDAC genes. As an HDACi, VPA interferes in particular with HDACs 1 and 2. HDACs have distinct patterns of expression in the embryo, with HDAC 1 and 2 being prominent in the early brain, thereby likely making cells in this region susceptible to effects of the drug [45–47]. Furthermore, although VPA is an HDACi, which are typically associated with gene activation, we find, as did others, that the developmental phenotypes are associated with the down-regulated and not the up-regulated genes [36]. This suggests that VPA induction of p19$^{Arf}$-mediated senescence causes a broad repression of key developmental pathways, which impact NE fate and contribute to the developmental phenotypes, as many of these were rescued in the absence of *p19$^{Arf}$*. Among these, we identify tRNA regulation as one of the most significantly restored pathways in the absence of *p19$^{Arf}$*. Importantly, p19$^{Arf}$ can directly block tRNA synthesis [48], while disruption of tRNA function is strongly associated with microcephaly and neurodevelopmental disorders [37,49–53]. Interestingly, recent findings also show that induction of senescence involves disruption of tRNA expression, further reinforcing this link [54]. It will be interesting to determine whether such inhibition of tRNA function contributes to specific, or global

alterations in protein translation in senescent cells, either in VPA-induced developmental defects, or other settings.

Overall, the discovery that atypical activation of senescence in the embryo can perturb development raises the intriguing possibility that it may also contribute to defects in developmental contexts beyond those we studied here and highlights how the study of mistimed senescence in developmental disorders merits further study.

## Materials and methods

### Animal maintenance and VPA administration

Pregnant CD1, C57Bl6/J, $p21$−/−, $p19^{Arf}$−/−, and $p16^{Ink4a}$−/− were maintained in a temperature- and humidity-controlled animal facility with a 12-hour light/dark cycle. We administrated 400 mg/kg VPA (Sigma-Aldrich, (Missouri, USA) P4543) or PBS as control, intraperitoneally to timed-pregnant females, at E8 (3 times (9 AM, 1 PM, and 4 PM). The $p21$ −/−, $p19^{Arf}$−/−, and $p16^{Ink4a}$ −/− mice were on a C57Bl6J background, so were compared to C57Bl6J wild-type as control. We observed that the C57Bl6J mice are more sensitive than the CD1 mice to induction of microcephaly. For qRT-PCR and RNA-seq analysis, only the first 2 doses were administered, and samples were collected at E8.75. All the experimental procedures were in full compliance with the institutional guidelines of the accredited IGBMC/ICS animal house, in compliance with French and EU regulations on the use of laboratory animals for research, under the supervision of Dr. Bill Keyes who holds animal experimentation authorizations from the French Ministry of Agriculture and Fisheries (#12840).

### Organoids

Cerebral organoids were generated from the iPSC line HPSI0214i-kucg_2 (Catalog# 77650065, HipSci) using the STEMdiff Cerebral Organoid Kit (Catalog nos. 08570 and 08571) from STEMCELL Technologies (Vancouver, Canada). Representative pictures were acquired with a LEICA DMS 1000. We acknowledge Wellcome Trust Sanger Institute as the source of HPSI0214i-kucg_2 human induced pluripotent cell line, which was generated under the Human Induced Pluripotent Stem Cell Initiative funded by a grant from the Wellcome Trust and Medical Research Council, supported by the Wellcome Trust (WT098051) and the NIHR/Wellcome Trust Clinical Research Facility, and acknowledges Life Science Technologies Corporation as the provider of "Cyto tune." Cultures were exposed to unbuffered VPA, diluted in medium. Analysis of the medium showed no pH change in response to VPA.

### Immunofluorescence

Embryos and organoids were fixed in 4% PFA for 30 minutes at 4˚C, washed in PBS, and processed for paraffin embedding. Sections were obtained using a microtome (8 μm, Leica 2035 Biocut). After antigen unmasking in citrate buffer (0.01 M, pH 6) for 15 minutes in a microwave oven, slides were blocked with 5% donkey serum, 0.1% TritonX-100 in PBS, and incubated overnight with the following primary antibodies: PHH3 (1:500, Upstate (Merck, Darmstadt, Germany) #05–806); Pax6 (1:300, Covance (New jersey, US) #PRB-278P); Tbr2 (1:300, eBioscience (Thermo Fisher Scientific, Massachusetts, USA) #14–4875); (1:300, Millipore (Massachusetts, USA), #AB2283); Sox1 (1:50, R&D Systems (Minnesota, USA) #AF3369); βIII-tubulin/Tuj1 (1:200, Covance #MMS-435P-100); $p19^{Arf}$ (5-C3-1) rat monoclonal antibody (Santa Cruz (Texas, USA) #sc-32748); and GFP 2A3 (IGBMC (Illkirch, France)). Primary antibodies were visualized by immunofluorescence using secondary antibodies from donkey (1:400, Invitrogen (California, USA): Alexa Fluor 568 donkey anti-mouse IgG #A-100037,

Alexa Fluor 488 donkey anti-rat IgG #A-21208, Alexa Fluor 488 donkey anti-rabbit #A-21206, Alexa Fluor 568 donkey anti-Goat IgG #A-11057) and from goat (1:400, Invitrogen: Alexa Fluor 568 goat anti-rabbit IgG #A-110111, Alexa Fluor 488 goat anti-mouse IgG #A11001, Alexa Fluor 568 anti-rat IgG #A11077), and cell nuclei were identified using DAPI (1:2,000). Stained sections were digitized using a slide scanner (Nanozoomer 2.0-HT, Hamamatsu, Japan), and measurements (thickness of the neuronal layer) were performed using the NDPview software of the digital scanner.

### SA-β-gal staining

Whole-mount SA-β-gal was detected as previously described [10]. Incubation with X-gal was performed overnight for mouse embryos and 1 hour and 30 minutes for organoids. For determination of specific localization of senescence in embryonic tissue, embryos stained with SA-β-gal were postfixed in 4% PFA overnight at 4˚ C, embedded in paraffin and sectioned. Representative pictures were acquired using a macroscope (Leica M420) and stained sections were digitized using a slide scanner (Nanozoomer 2.0-HT, Hamamatsu).

### EdU

To assess cell proliferation in embryos, pregnant female mice at E9.5 were injected intraperitoneally with 5-ethynyl-2′-deoxyuridine (EdU; 50 mg/kg body weight) for 1 hour. Click-iT EdU Alexa Fluor 488 Imaging Kit (Thermo Fisher (Massachusetts, USA)) was used as per manufacturer's protocol. Representative pictures were acquired using a microscope (DM4000B).

### TUNEL

Cell death was assessed using the TdT-mediated dUTP nick end-labeling (TUNEL) method (ApopTagPeroxidase In Situ Apoptosis detection kit, Millipore) as per manufacturer's instructions. Representative pictures were acquired using a macroscope (Leica M420) and a microscope (DM4000B).

### RT-qPCR and analysis

The combined forebrain and midbrain region was manually dissected from E8.75 embryos, and snap-frozen. RNA was extracted from individual embryos using the RNAeasy mini kit (QIAGEN (Hilden, Germany)). Moreover, 10 ng RNA were used for analysis with the LUNA one-step RT-qPCR kit (LUNA E3005L BioLabs (Massachusetts, USA)). The relative expression levels of the mRNA of interest were determined by real-time PCR using Quantifast SYBR Green Mix (QIAGEN) with specific primers listed in S1 Table and a LightCycler 480 (Roche (Basel, Switzerland)). Samples were run in triplicate and gene of interest expression was normalized to human Gapdh or mouse Rplp0.

### In ovo electroporation

Fertilized chicken embryos were obtained from local farmers. Chick eggs were incubated in a humidified chamber at 37˚C. Moreover, 1.5 μg/μL DNA constructs (*pCAGGS-GFP* [a gift from Dr. J. Godin, IGBMC] or *pCAGGS-p19^{ARF}-GFP* [*p19^{Arf}* coding sequence was cloned in XhoI/NheI multiple cloning sites in the *pCAGGS-GFP*]) mixed with 0.05% Fast Green (Sigma-Aldrich, (Missouri, USA)) were injected into neural tubes of stage HH8 chick embryos and electroporated on the right side, leaving the left side as untreated control. Electroporation was performed using a square wave electroporator (BTX ECM 830 electroporation system) and the parameters applied: 3 pulses of 15V for 30 ms with an interval of 1 second. Embryos were

harvested 24 hours after electroporation and processed for SA-β-gal, histology and immuno-histochemistry. Representative pictures were acquired using a macroscope (Leica Z16 APO) and a microscope (Leica DM4000B).

## Whole-mount in situ hybridization

RNA probes were prepared by in vitro transcription using the Digoxigenin-RNA labeling mix (Roche). Template plasmids were kindly provided by Drs G. Oliver (*Six3*) and S.L. Ang (*Mox1*). Mouse embryos were dissected in ice-cold PBS and fixed O/N in 4% PFA/PBS. After several washes in PBS1X/0.1% Tween-20 (PBT), embryos were bleached for 1 hour in 3% $H_2O_2$/PBT and washed in PBT before being digested with Proteinase K (10mg/ml) for 2 minutes. Digestion was stopped by 5-minute incubation in 2 mg/ml glycine/PBT. Embryos were washed again in PBT before postfixing for 20 minutes in 0.2% glutaraldehyde/4% PFA/PBS. After further washes they were incubated in prewarmed hybridization buffer (50% deionized formamide, 5XSSC, 1%SDS, 100μg/ml tRNA) and prehybridized for 2 hours at 65°C. The buffer was then replaced with fresh prewarmed hybridization buffer containing the digoxigenin labeled RNA probes and incubated O/N at 65°C. The next day, embryos were washed twice in buffer 1 (50% formamide; 5XSSC; 1%SDS) at 65°C then in buffer 2 (NaCl 500mM, 10mM TrisHCl pH = 7.5, 0.1%Tween20) at room temperature before treating them with RNaseA (100 mg/ml) to reduce background. The embryos were rinsed in buffer 2, then in buffer 3 (50% formamide, 2XSSC). Finally, the embryos are rinsed in TBS/0.1% Tween-20 (TBST) then blocked for 2 hours in 2% blocking solution (Roche) and incubated O/N in the same solution containing 1:2,500 anti-digoxigenin antibody (Roche). The next day, the embryos were washed in TBST, before washing them in NTMT (NaCl 100mM, Tris-HCl 100mM pH = 9,5, $MgCl_2$ 50mM, Tween20 at 0.1%) and developing the signal in the dark with staining solution (4.5 μl/ml NBT and 3.5 μl/ml BCIP (Roche) in NTMT buffer).

## RNA sequencing

RNA was collected as for qRT-PCR. Full-length cDNA was generated from 10 ng of total RNA from 4 individual embryos per treatment, using Clontech SMART-Seq v4 Ultra Low Input RNA kit for Sequencing (Takara Bio Europe, Saint Germain en Laye, France) according to the manufacturer's instructions with 8 cycles of PCR for cDNA amplification by Seq-Amp polymerase. A total of 600 pg of preamplified cDNA were then used as input for Tn5 transposon tagmentation by the Nextera XT DNA Library Preparation Kit (96 samples) (Illumina, San Diego, California, USA) followed by 12 cycles of library amplification. Following purification with Agencourt AMPure XP beads (Beckman-Coulter, Villepinte, France), the size and concentration of libraries were assessed by capillary electrophoresis using the Agilent 2100 Bioanalyzer.

Sequencing was performed on an Illumina HiSeq 4000 in a 1x50bp single end format. Reads were preprocessed using cutadapt 1.10 in order to remove adaptors and low-quality sequences, and reads shorter than 40 bp were removed from further analysis. Remaining reads were mapped to Homo sapiens rRNA sequences using bowtie 2.2.8, and reads mapped to those sequences were removed from further analysis. Remaining reads were aligned to mm10 assembly of Mus musculus with STAR 2.5.3a. Gene quantification was performed with htseq-count 0.6.1p1, using "union" mode and Ensembl 101 annotations. Differential gene expression analysis was performed using DESeq2 1.16.1 Bioconductor R package on previously obtained counts (with default options). *p*-Values were adjusted for multiple testing using the Benjamini and Hochberg method. Adjusted *p*-value $<0.05$ was taken as statistically significant.

Pathway analysis was performed using Enrichr (http://amp.pharm.mssm.edu/Enrichr) with Gene Ontology 2018 and MGI Mammalian Phenotype Level 4 2019 databases. Adjusted *p*-value of <0.25 was used as a threshold to select the significant enrichment (Fig 6A, all terms significant in WT, with only "exencephaly," significantly enriched in p19KO (adj. <0.25): Fig 6C, all terms significant (adj. <0.25): S12 Fig, all terms significant in WT, and not in p19KO (adj. <0.25)). RNA sequencing data are available at GEO (GSE175680). All other relevant data are within the paper.

## Counting and statistical analysis

For cell number quantification, positive cells for a given marker (Pax6, Sox1, Tbr2, Tuj1, and PHH3) were counted in a 100-μm wide columnar area from the VZ to the apical surface in similar regions in the cortex. Edu was counted similarly, in a 50-μm wide columnar area. Immunofluorescence analyses, area measurements, and RNA expression were statistically analyzed using Prism (GraphPad, San Diego, California, USA). At least 5 animals of each treatment from 3 different litters were analyzed. Cell counting was performed on 3 adjacent sections. Results are presented as mean ± SEM. Statistical analysis was carried out employing the Mann–Whitney test for unpaired variables. For 3 or more groups, normal multiple comparisons were tested with 1-way ANOVA plus Tukey post hoc test and nonnormal multiple comparisons were tested using Kruskal–Wallis test followed by a Dunn test. *p*-Values < 0.05 were considered significant ($^*p \leq 0.05$, $^{**}p \leq 0.01$, $^{***}p \leq 0.001$, and $^{****}p \leq 0.0001$).

## Supporting information

**S1 Fig. VPA treatment affects somite number and embryo length. (A)** Quantification of visibly intact somite number (Control, *n* = 27 from 12 litters; (OB), Open brain, *n* = 22 from 10 litters; (SB) Small brain, *n* = 25 from 16 litters). Data bars represent mean ± SEM. Kruskal–Wallis test: ns, no significant and $^{***}p \leq 0.001$. **(B)** Measurements of the length of the embryo (from the otic vesicle to the tail tip) (Control, *n* = 24 from 12 litters; Open brain, *n* = 22 from 10 litters; Small brain, *n* = 25 from 16 litters. Data bars represent mean ± SEM. Kruskal–Wallis test: ns, no significant and $^{****}p \leq 0.0001$. The data underlying this figure can be found in S1 Data. VPA, valproic acid.
(TIF)

**S2 Fig. VPA decreases cell proliferation. (A)** Quantification of total EdU positive cells present at E9.5 (Control, n = 9 from 5 litters; Open brain, *n* = 4 from 3 litters; Small brain, *n* = 5 from 3 litters). Data bars represent mean ± SEM. Kruskal–Wallis test: $^*p \leq 0.05$. **(B)** Quantification of EdU positive cells in the apical zone at E9.5 (Control, *n* = 9 from 5 litters; Open brain, *n* = 4 from 3 litters; Small brain, *n* = 5 from 3 litters). Data bars represent mean ± SEM. Kruskal–Wallis test: $^*p \leq 0.05$ and $^{**}p \leq 0.01$. **(C)** Left: Immunostaining on sections of control and VPA-treated embryos for PHH3 (red) at E9.5. The square indicates the counted area. Scale bar, 100 μm. Right: PHH3 positive cells quantification at E9.5. (Control, *n* = 4 from 2 litters; Open brain, *n* = 3 from 3 litters; Small brain, *n* = 4 from 3 litters from 3 litters); 3 levels have been counted per embryo. Data bars represent mean ± SEM. Kruskal–Wallis test: $^{**}p \leq 0.01$ and $^{****}p \leq 0.0001$. The data underlying this figure can be found in S1 Data. E, embryonic day; PHH3, phospho-histone H3; VPA, valproic acid.
(TIF)

**S3 Fig. VPA does not induce apoptosis in the forebrain neuroepithelium.** Control and VPA-treated embryos were stained with whole mount TUNEL assay, to assess cell death. (Left) Lateral views and frontal views of control and VPA-treated embryos dissected at E8.5. Scale

bar, 500 μm. Corresponding horizontal sections at the forebrain level (3 embryos from at least 2 litters were analyzed). Scale bar, 100 μm. (Right) Lateral views and frontal views of control and VPA-treated embryos dissected at E9.5 (6 embryos from at least 5 litters were analyzed). Scale bar, 500 μm. Corresponding horizontal sections at the forebrain level. Scale bar, 100 μm. Some apoptotic cells are observed in the surface ectoderm. Positive cells are seen in the neural fold tips in all conditions (asterisk). E, embryonic day; ne, neuroepithelium; se, surface ectoderm; VPA, valproic acid.
(TIF)

**S4 Fig. Impaired neurogenesis after VPA treatment is already observed at E10.5.** Cortical sections (coronal) of E10.5 embryos were immunoassayed for Pax6, Tbr2, Tuj1, and counterstained with Dapi. Scale bar, 250 μm (top row), 50 μm. Graphs show quantification of Pax6 and Tbr2 positive progenitors or the thickness of the neuronal layer in the microcephalic cortical vesicles (5 embryos from at least 4 different mothers were analyzed). Data bars represent mean ± SEM Mann–Whitney test: $^*p \leq 0.05$ and $^{**}p \leq 0.01$. The data underlying this figure can be found in S1 Data. E, embryonic day; VPA, valproic acid.
(TIF)

**S5 Fig. Neurogenesis is impaired in human cerebral organoids treated with VPA.** Sections through control and VPA-treated organoids were immunostained with Sox1(red), Tbr2 (green), and Dapi (blue) at day 42 (scale bar, 50 μm), ($n$ = 15 (Control), 12 (1 mM VPA), 13 (2 mM VPA), 4 independent experiments). Kruskal–Wallis test: $^{***}p \leq 0.001$ and $^{****}p \leq 0.0001$. The data underlying this figure can be found in S1 Data. VPA, valproic acid.
(TIF)

**S6 Fig. Genetic deficiency of $p16^{Ink4a}$ or $p21$ does not rescue VPA-induced phenotypes.** Lateral views of control and VPA-treated embryos deficient for $p16^{Ink4a}$ or $p21$ (top row). Scale bar, 500 μm. Higher magnification of the heads in lateral (middle row) and frontal views (bottom row). Scale bar, 500 μm. An open neural tube or a smaller brain, as well as a gross misalignment of the neural tube and somites are still observed after VPA treatment in the absence of $p16^{Ink4a}$ or $p21$. VPA, valproic acid.
(TIF)

**S7 Fig. The impaired somite number and reduced length is not rescued in $p19^{Arf}$-deficient mice after VPA treatment. (A)** Quantification of visibly intact somite number (WT, $n$ = 8 embryos from 4 litters, WT+VPA, $n$ = 30 embryos from 11 litters, p19KO, $n$ = 14 embryos from 4 litters, p19KO + VPA, $n$ = 17 embryos from 8 litters). Data bars represent mean ± SEM. Kruskal–Wallis test: ns, no significant, $^*p \leq 0.05$ and $^{****}p \leq 0.0001$. **(B)** Measurements of the length of the embryo (from the otic vesicle to the tail tip (WT, $n$ = 8 embryos from 4 litters, WT+VPA, $n$ = 28 embryos from 11 litters, p19KO, $n$ = 14 embryos from 4 litters, p19KO + VPA, $n$ = 22 embryos from 8 litters). Data bars represent mean ± SEM. Kruskal–Wallis test: ns, not significant and $^{****}p \leq 0.000$. **(C)** Lateral views (top) and dorsal views (bottom) of control and VPA-treated embryos dissected at E9.5, illustrating the pronounced curve in the neural tube and abnormally shaped somites observed (control embryo is same as shown in S6 Fig). Scale bar, 500 μm. The data underlying this figure can be found in S1 Data. E, embryonic day; VPA, valproic acid; WT, wild-type.
(TIF)

**S8 Fig. Improved forebrain phenotype in $p19^{Arf}$-deficient mice after VPA treatment.** Whole mount in situ hybridization for *Six3* (forebrain) and *Mox1* (somites), showing an increased size of the forebrain in $p19^{Arf}$-deficient, VPA-treated mice in comparison to the *WT*

mice treated with VPA. Scale bar, 500 μm (top row) and 50 μm (bottom row). The number of embryos examined are indicated ($n = 10$ from at least 5 different litters). VPA, valproic acid; WT, wild-type.
(TIF)

**S9 Fig. Senescence is induced in the forebrain neuroepithelium in *p16^Ink4a*- or *p21*-deficient mice treated with VPA.** Whole mount SA-β-gal staining in control and VPA-treated embryos with small-brain phenotypes at E9.5 (WT, $n = 9$ embryos from 4 litters, WT+VPA, $n = 10$ embryos from 5 litters, p16KO, $n = 2$ embryos from 1 litter, p16KO + VPA, $n = 10$ embryos from 5 litters, p21KO, $n = 6$ embryos from 3 litters, p21KO + VPA, $n = 5$ embryos from 2 litters). Scale bar, 500 μm. Higher magnification of the heads in lateral (second row) and frontal views (third row). Scale bar, 50 μm. Bottom row, Sections through whole mount SA-β-gal stained forebrains (scale bar, 100 μm). Red asterisks highlight senescent cells. (WT, $n = 5$ embryos from 3 litters, WT+VPA, $n = 10$ embryos from 5 litters, p16KO, $n = 2$ embryos from 1 litter, p16KO + VPA, $n = 10$ embryos from 5 litters, p21KO, $n = 6$ embryos from 3 litters, p21KO + VPA, $n = 5$ embryos from 2 litters). E, embryonic day; WT, wild-type.
(TIF)

**S10 Fig. Senescence and SASP genes are less induced in *p19^Arf*-deficient embryos with VPA treatment.** qRT-PCR analysis on E8.75 forebrain and midbrain, from control and *p19^Arf*-deficient mice, treated with VPA or left untreated. Graphs show fold change expression for the senescence markers (*p21* and *p16^Ink4a*) and for SASP genes (*IL6*, *IL1a*, *IL1b*, and *Pai1*), normalized to untreated control ($n = 12$ (Control), $n = 12$ (Control+VPA), $n = 20$ (*p19KO*), $n = 20$ (*p19KO*+VPA), from at least 3 different litters). Data bars represent mean ± SEM. Kruskal–Wallis test: ns, no significant, ${}^{**}p \leq 0.01$, ${}^{***}p \leq 0.001$ and ${}^{****}p \leq 0.0001$. The data underlying this figure can be found in S1 Data. E, embryonic day; qRT-PCR, quantitative real-time PCR; SASP, senescence-associated secretory phenotype; VPA, valproic acid.
(TIF)

**S11 Fig. Ectopic expression of *p19^Arf*-GFP in the chicken neural tube.** Sections through the neural tube of *p19^Arf*-GFP electroporated chicken embryos electroporated at stage HH12, immunostained for p19^Arf (red) and GFP (green), with Dapi counterstaining (blue). The green star shows the electroporated side. Scale bar, 100 μm.
(TIF)

**S12 Fig. RNA-seq data analysis uncovers neurodevelopmental and tRNA-related signatures as being less affected by VPA treatment in *p19^Arf*-deficient mice. (A)** GO Biological Processes pathway analysis on the down-regulated genes from RNA-seq of the forebrain and midbrain. Heat maps showing the relative expression of representative genes associated with **(B)** microcephaly (list generated from [35]), **(C)** autism (list generated from [35]) and **(D)** tRNA (list of genes identified in Fig 6C pathway analysis). The data underlying this figure can be found in S1 Data. RNA-seq, RNA sequencing; VPA, valproic acid.
(TIF)

**S1 Table. Primers used for qRT-PCR in the study.** qRT-PCR, quantitative real-time PCR.
(DOCX)

**S1 Data. Excel spreadsheet containing, in separate sheets, the underlying numerical data for Figs** 2D, 3, 4B, 4C, 4E, 4F, 5A, 5E, 5F, 6A, 6C, S1A, S1B, S2A, S2B, S2C, S4, S5A, S7A, S7B, S10, S12A, S12B, S12C and S12D.
(XLSX)

## Acknowledgments

We thank Travis Stracker, Juliette Godin, Michele Studer, Pura Munoz, Birgit Ritschka, Christina Lilliehook (Life Science Editors), and members of the Keyes lab for comments on the manuscript. We thank the core facilities at the IGBMC for excellent technical support, including the Sequencing, Cell Culture and Microscopy platforms, the mouse facilities of the IGBMC and the Mouse Clinical Institute (ICS), and in particular Sylvie Falcone, Amélie Freismuth, Marion Humbert, Jean-Marie Garnier, and Olivia Wendling for technical assistance.

## Author Contributions

**Conceptualization:** Muriel Rhinn, William M. Keyes.

**Data curation:** Muriel Rhinn, Irene Zapata-Bodalo.

**Formal analysis:** Muriel Rhinn, Irene Zapata-Bodalo, Annabelle Klein.

**Funding acquisition:** William M. Keyes.

**Investigation:** Muriel Rhinn, Annabelle Klein, Jean-Luc Plassat, Tania Knauer-Meyer, William M. Keyes.

**Methodology:** Muriel Rhinn, Irene Zapata-Bodalo, Annabelle Klein, Jean-Luc Plassat, Tania Knauer-Meyer.

**Project administration:** Muriel Rhinn, William M. Keyes.

**Supervision:** William M. Keyes.

**Validation:** Muriel Rhinn, Annabelle Klein, Jean-Luc Plassat.

**Visualization:** Muriel Rhinn, Irene Zapata-Bodalo.

**Writing – original draft:** Muriel Rhinn, William M. Keyes.

**Writing – review & editing:** Muriel Rhinn, Irene Zapata-Bodalo, Annabelle Klein, Jean-Luc Plassat, Tania Knauer-Meyer.

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
