## [Editor Report · Decision Letter 0]

4 Nov 2021

Dear Dr Keyes, 

Thank you for submitting your manuscript entitled "Aberrant induction of p19Arf-mediated cellular senescence contributes to neurodevelopmental defects" for consideration as a Research Article by PLOS Biology.

Your manuscript has now been evaluated by the PLOS Biology editorial staff as well as by an academic editor with relevant expertise and I am writing to let you know that we would like to send your submission out for external peer review.

Once your full submission is complete, your paper will undergo a series of checks in preparation for peer review. Once your manuscript has passed the checks it will be sent out for review. 

Please re-submit your manuscript within two working days, i.e. by Nov 08 2021 11:59PM.

Given the disruptions resulting from the ongoing COVID-19 pandemic, please expect delays in the editorial process. We apologize in advance for any inconvenience caused and will do our best to minimize impact as far as possible.

Kind regards,

Lucas

Lucas Smith

Associate Editor

PLOS Biology

lsmith@plos.org

P.S. As a final note: If your manuscript has been previously reviewed at another journal, PLOS Biology is willing to work with those reviews in order to avoid re-starting the process. Submission of the previous reviews is entirely optional and our ability to use them effectively will depend on the willingness of the previous journal to confirm the content of the reports and share the reviewer identities. Please note that we reserve the right to invite additional reviewers if we consider that additional/independent reviewers are needed, although we aim to avoid this as far as possible. In our experience, working with previous reviews does save time. 

If you would like to send your previous reviewer reports to us, please specify this in the cover letter, mentioning the name of the previous journal and the manuscript ID the study was given, and include a point-by-point response to reviewers that details how you have or plan to address the reviewers' concerns. Please contact me at the email that can be found below my signature if you have questions.

---

## [Decision Letter · Decision Letter 1]

6 Dec 2021

Dear Dr Keyes,

Thank you for submitting your manuscript "Aberrant induction of p19Arf-mediated cellular senescence contributes to neurodevelopmental defects" for consideration as a Research Article at PLOS Biology. Your manuscript has been evaluated by the PLOS Biology editors, an Academic Editor with relevant expertise, and by several independent reviewers.

The reviews of your manuscript are appended below. As you will see from their comments, the reviewers are generally positive about the study, and Reviewers 1 and 3 have relatively minor requests. However, Reviewer 2 has identified a number of important concerns which undermine some of the conclusions presented here, and s/he highlights the need for additional analyses to strengthen the study. After discussion with the Academic Editor, we feel that it would be important to address Reviewer 2's concerns with new data and analyses where appropriate.

In light of the reviews, we will not be able to accept the current version of the manuscript, but we would welcome re-submission of a much-revised version that thoroughly addresses the reviewers' comments. We cannot make any decision about publication until we have seen the revised manuscript and your response to the reviewers' comments. Your revised manuscript is also likely to be sent for further evaluation by the reviewers.

We expect to receive your revised manuscript within 3 months. 

**IMPORTANT - SUBMITTING YOUR REVISION**

*Re-submission Checklist*

*Published Peer Review*

*PLOS Data Policy*

*Blot and Gel Data Policy*

Sincerely,

Lucas Smith

Associate Editor

PLOS Biology

lsmith@plos.org

REVIEWS:

Reviewer #1: Rhinn et al test whether VPA-induced developmental defects involves senescence. They characterise the embryonic phenotype in mice, showing the VPA treatment induces malformation such as exencephaly, microcephaly and spinal defects, as well as defective neurogenesis in those affected tissues. They further show that VPA induces ectopic senescence in neuroepithelial cells during developmental neurogenesis. Genetically, they show evidence that p19 is sufficient and partially necessary for microcephaly induction. The data quality seems high and senescence assays are thorough. This is an interesting and highly original study, reinforcing the critical role of senescence in embryonic development. 

I only have a few rather minor points as follows.

They find no improvement in the VPA-phenotype in p21- or p16-defective embryos. How about the senescence phenotype? Do these embryos still show VPA-associated senescence or do the developmental defects appear despite senescence reduction?

This might be beyond the scope and results may not be decisive, but I am curious if they see increased histone acetylation in VPA-treated embryos and/or organoids. Do they see a similar effect with other HDACi in organoids? 

Fig. 5 and 7 could be better integrated? 

Reviewer #2: In this substantial report, Rhinn and colleagues use transgenic mice, electroporated chick embryos and human cerebral organoids to convincingly demonstrate that the anti-epileptic drug VPA causes p19ARf-mediated senescence of neuroepithelial cells. The comments below are intended to clarify aspects of the phenotyping and mechanistic links between induction of senescence and phenotypes observed.

1) The decreasing proportion of microcephalic embryos observed at E9.5-E13.5 in Fig 1B is perplexing but may be caused by misattribution at earlier stages. It is common for embryos with small open hindbrain or anterior neuropores to present with a smaller head due to lack of expansion of the presumptive ventricles. In many models, small cranial failures of neural tube closure expand as the tissue grow and become evident exencephaly at later stages. Misattribution is evidenced in the embryo labelled "Microcephaly" in Fig 2A, which has a midbrain opening visible in the images provided. The authors are encouraged to remove the phenotype data at E9.5/E10.5, leaving the more definitive E13.5 data. Quantitative analysis of head size relative to embryo body size is also essential to validate the microcephaly phenotype rather than a general stunting of growth. 

2) Please use the E9.5/E10.5 data to provide standard measures of embryo development such as somite number, dorsal length, turning score, etc. The authors may also wish to comment on secondary phenotypes such as hypoplastic pharyngeal arches visible in Fig 1B, which are also potentially relevant to fetal valproate syndrome. Note that cranial neural tube closure is not reliable completed before the ~17 somite stage in C57Bl/6J embryos so only embryos with >17 somites should be assessed for this phenotype (e.g. the WT control in Fig S4 with an open cranial NT has fewer than 17 somite, as does the p16KO labelled as Microcephaly in the same figure but which actually has an open cranial NT clearly visible at the apex of the head).

3) The concentrations of valproic acid used in culture are rather high. No information is given on whether this substance was buffered prior to addition to culture (please also indicate the catalogue number of the product purchased as Sigma offer various formulations). What was the pH of the culture medium after addition of 2 mM VPA?

4) It is convincing that the VPA-exposed organoids are smaller. It is therefore not surprising that the neural rosette surface and Tuj1 thickness are smaller in the treated organoids. Please normalise these to a measure of organoid size to clarify whether VPA largely acted to restrict neuroepithelial cell expansion (as suggested by other data in the manuscript) or also subsequently impaired neural differentiation.

5) Fig 5A: The apparent exacerbation of exe induced by VPA in p21 and p16 KO is striking. Non-VPA treated controls need to be provided to support interpretation of this data. This data is limited because it was performed at E9.5 when the distinction between exe and microcephaly is in doubt (comment 1). Nonetheless it may suggest that p21/p16 expression is protective, enabling the embryo to convert the fatal exe phenotype into viable microcephaly. In this E9.5 cohort, were other features of embryo development such as somite gain also rescued by p19 KO or was the effect limited to the cranial neural tube?

6) The apical localisation of B-gal stained cells in the neuroepithelium in vivo is striking. Can the authors comment on the distribution of these cells? Restricted apical distribution could be an artefact if the embryos were stained in wholemount before sectioning (as described in the methods) due to limited penetration of B-gal. This is an important consideration given the exposed NE or smaller heads of the VPA-treated embryos will enable greater reagent access (this reviewer is convinced that senescence really is increased, but potentially not to the extent and localisation suggested by the images).

7) SOX1/apical area is quantified in the organoid system, whereas Pax6 and Tbr2 are quantified in mice. Can the same parameters be provided in both systems?

8) Does the RNASeq dataset provide any insights into whether it is the presence of senescent cells, or the non-senescent cells' responses to SASP which underlies the phenotypes observed?

9) Fig 7: Were only microcephalic embryos included in the WT+VPA group, or did this include exe embryos with degenerating NE?

10) The authors interpret the EdU assay as showing NE cells are "proliferative in control but not in VPA-exposed embryonic mice." This interpretation is not clear given the extensive number of EdU-positive, basally-located NE cells in both groups (Fig 2 C). EdU should label cells with basally-located nuclei as this is where S phase occurs. In the images provided it appears that in the control embryo some cells had progressed through the cell cycle whereas in the VPA-treated embryos most are still basal. Please quantify the EdU staining to support interpretation.

11) Similarly, the TUNEL staining is unconvincing without quantification. Although the authors interpret the images as showing increased positivity in the non-neural ectoderm, staining is also evident in the neural fold tips.

12) Were all organoid studies performed with a single cell line? 

Reviewer #3: The manuscript by Rhinn and colleagues illustrates a novel role for cellular senescence in the context of embryonic development. The paper describes a new role for p19/p14 during development, which is detrimental for brain development. 

This is a high quality paper, very well written and experiments are well conducted. Conclusions are fully supported by the data. 

I believe this paper should be published-I have no major suggestions to improve it. 

1 minor comment: Figure 6 in C it says genes highlighted in E, but I guess the authors mean B.

---

## [Editor Report · Decision Letter 2]

11 Apr 2022

Dear Dr Keyes,

Thank you for submitting your manuscript "Aberrant induction of p19Arf-mediated cellular senescence contributes to neurodevelopmental defects" for consideration as a Research Article by PLOS Biology. Your resubmission was evaluated by the PLOS Biology editors as well as by an Academic Editor with relevant expertise. In this case the Academic Editor felt comfortable evaluating your revisions so it was not sent back out to the original reviewers.

Based on feedback from the Academic Editor, we will probably accept this manuscript for publication, provided you satisfactorily address the remaining points below. Please also make sure to address the data and other policy-related requests at the bottom of this email that are required for final acceptance.

1) We'd suggest a modification to your abstract that flips it around to highlight the novelty somewhat more. A draft of what we're suggesting is:

Valproic-acid (VPA) is a widely prescribed drug to treat epilepsy, bipolar disorder and migraine. If taken during pregnancy however, exposure to the developing embryo can cause birth defects, cognitive impairment and Autism-Spectrum Disorder. How VPA causes these developmental defects remains unknown. We used embryonic mice and human organoids to model key features of VPA drug exposure, including exencephaly, microcephaly and spinal defects. In the malformed tissues, in which neurogenesis is defective, we find pronounced induction of cellular senescence in the neuroepithelial cells. Critically, through genetic and functional studies, we identified p19 Arf as the instrumental mediator of senescence and microcephaly, but surprisingly, not exencephaly and spinal defects. Together, these findings demonstrate that misregulated senescence in neuroepithelial cells can contribute to developmental defects.

2) Please add subheaders to the Results section to give better flow to the work.

3) Please combine Fig 5 and 7 (as per an original reviewer request). We agree with the reviewer that this helps with the flow of the study. Please also adjust the manuscript text accordingly.

4) Finally, while we were convinced by your arguments for doing your analyses the way that you did, we ask that you provide rational in the final version of the paper explaining why you chose to do the analyses the way you did and not the way that Reviewer 2 asked for. We feel this is important as similar questions may be raised by our readership.

We expect to receive your revised manuscript within two weeks. 

*Published Peer Review History*

*Press*

Sincerely,

Kris

Kris Dickson,

Neurosciences Senior Editor/Section Manager,

kdickson@plos.org,

PLOS Biology

DATA POLICY:

***In this case, please ensure that the RNA seq data deposited to the NCBI Gene Expression Omnibus (“Accession "GSE175680") is released now, as that information will need to be made available prior to final acceptance. It is currently private and is listed as being scheduled to be released on May 27, 2022.

Regardless of the method selected, please ensure that you provide the individual numerical values that underlie the summary data displayed in the relevant figure panels as they are essential for readers to assess your analysis and to reproduce it. This includes the underlying data used to produce the graphs and heat maps in figures: 

Fig2D; Fig3graphs; Fig4B-Egraphs,F; Fig5A,E, Fig6A-C, Fig7 graphs, SFig1A,B; SFig2A-C, SFig4graphs; SFig5graph; Sfig7A,B; SFig10; SFig12A-D

***NOTE: the numerical data provided should include all replicates AND the way in which the plotted mean and errors were derived (it should not present only the mean/average values).***

***Please also ENSURE THAT IN EACH FIGURE LEGEND (both main and supplementary) you include information on WHERE THE UNDERLYING DATA CAN BE FOUND, and ensure your supplemental data file/s has a legend.

DATA NOT SHOWN?

---

## [Editor Report · Decision Letter 3]

6 May 2022

Dear Bill,

On behalf of my colleagues and the Academic Editor, Judith Campisi, I am pleased to say that we can in principle accept your Research Article "Aberrant induction of p19Arf-mediated cellular senescence contributes to neurodevelopmental defects" for publication in PLOS Biology, provided you address any remaining formatting and reporting issues. These will be detailed in an email that will follow this letter and that you will usually receive within 2-3 business days, during which time no action is required from you. Please note that we will not be able to formally accept your manuscript and schedule it for publication until you have completed any requested changes.

PRESS

We frequently collaborate with press offices. If your institution or institutions have a press office, please notify them about your upcoming paper at this point, to enable them to help maximize its impact. If the press office is planning to promote your findings, we would be grateful if they could coordinate with biologypress@plos.org. If you have previously opted in to the early version process, we ask that you notify us immediately of any press plans so that we may opt out on your behalf.

Sincerely, 

Kris

Kris Dickson, Ph.D. 

Senior Editor 

PLOS Biology

kdickson@plos.org